# Analysis of Spatiotemporal Interaction Characteristics and Decoupling Effects of Urban Expansion in the Central Plains Urban Agglomeration

**Zhaolan Wang [1,2], Li Wang [1], Biao Zhao [3,*] and Qian Pei [1]**

[1] School of Geography, Liaoning Normal University, Dalian 116029, China; wangzhaolan922@163.com (Z.W.); wangli@lnnu.edu.cn (L.W.); peiqian6786@163.com (Q.P.)

[2] Department of History and Geography, Tonghua Normal University, Tonghua 134001, China

[3] Institute of Chinese Borderland Studies, CASS, Beijing 100101, China

**\*** Correspondence: zhaobiao@igsnrr.ac.cn

**Abstract:** In recent years, the rapid improvement in the urbanization level of the Central Plains urban agglomeration is bound to bring about significant changes in urban land expansion and economic development. However, at present, there is little attention paid to the research on the spatiotemporal interaction characteristics of urban expansion and the interaction between urban expansion and economic development in this region, and existing research lacks a geographical analysis perspective. This study uses spatial autocorrelation, hot spot analysis, LISA time path, and standard deviation ellipse models to analyze the spatiotemporal interaction characteristics of urban expansion in the Central Plains urban agglomeration from 1990 to 2020, and it uses bilateral spatial autocorrelation and decoupling models to analyze the spatial correlation and decoupling effects of urban expansion and economic development. The results show that (1) the urban built-up area of the Central Plains urban agglomeration as a whole is growing in a "J" shape, and the expansion rate has increased rapidly in the past 10 years. (2) The spatial expansion of the city is mainly in the direction of "northwest–southeast"; the directionality has been gradually strengthened in the past 10 years, mainly in the direction of several prefecture-level cities under the jurisdiction of Anhui Province, and the spatial center of gravity of the city has shifted significantly to the south. (3) The spatial agglomeration characteristics of urban expansion in the Central Plains urban agglomeration are not obvious; local hot spots are concentrated in Jiaozuo and its surrounding areas, and urban expansion has local spatial structural instability. (4) During the 2005–2020 period, the risk of uncoordinated urban expansion and economic growth in the Central Plains urban agglomeration increased. This study is of great significance for the rational control of regional development, providing empirical reference for the formulation of the development planning of the Central Plains urban agglomeration, as well as providing a reference for research ideas and methods related to urbanization.

**Keywords:** urban expansion; spatiotemporal interaction; decoupling effect; Central Plains urban agglomeration

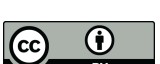

## 1. Introduction

The improvement of urbanization level is the main driving force for regional economic development [1], and it is also an important symbol of population non-agriculturalization and the progress of human civilization. According to the "World Cities Report 2020", the world will further urbanize in the next decade, and the proportion of global urban population is expected to reach 60.4% by 2030 [2]; population urbanization is bound to promote the further development of urban expansion [3]. China is in a period of rapid urbanization, with the disorderly expansion of cities causing waste of land resources [4,5], with some cities experiencing aggressive "incremental" expansion, and the phenomenon of "ghost cities" has emerged. Extensive urban expansion can exacerbate urban displacement

and affect quality of life [6]. Relevant research indicators should pay more attention to the improvement of housing quality, as well as the improvement of urban housing quantity [7]. With the development of technology, especially the development of the digital economy, the economic growth of cities is not at the cost of consuming land resources. Urban expansion is the most intuitive manifestation of urbanization. Therefore, the exploration of the characteristics of urban expansion and the analysis of the relationship between urban expansion and economic growth have become research fields of concern to scholars.

At present, there are many related studies on urban expansion in academic circles. From the perspective of the research content, it mainly includes the following aspects: First, there is research on urban expansion based on mathematical statistics. The commonly used evaluation perspectives include urban expansion intensity, expansion speed, expansion direction, urban compactness, etc. [8–10]. Second, there are studies about the spatiotemporal pattern characteristics of urban expansion: more scholars analyze the spatial agglomeration characteristics of urban expansion and landscape pattern characteristics, and the common methods include spatial statistical analysis and landscape pattern index in landscape ecology [11,12]. Third is the study of drivers of urban expansion [13,14], and fourth is the study of simulation and prediction of urban expansion [15–17], in which meta-automata are often used to simulate urban expansion. Fifth, there is the study of the interaction between urban expansion and transportation, ecological environment, and peri-urban agriculture [18–20]. Statistical data, remote sensing image interpretation data, and nighttime lighting data are often used to calculate the area of urban built-up areas in different periods [9,13,21]. The above research contents, methods and data sources can reflect the spatiotemporal characteristics of urban expansion and its impacts from different aspects, which have important academic values. However, the analysis of spatiotemporal patterns of urban expansion mostly separates time and space, ignoring the study of the spatial and temporal interaction characteristics of the urban expansion process.

Urbanization in China has led to accelerated urban expansion, and urban expansion is significantly correlated with gross domestic product (GDP) per capita [22]. Studies have shown that economic development affects the process of urban expansion, but in the long run, economic development also inhibits urban expansion, and urbanization and economic development have a positive impact on environmental degradation [23,24]. The growth of China's urban economy is influenced by the land factor [25], and moderate urban sprawl has a significant positive impact on economic development, while excessive urban sprawl has a significant negative impact on economic development [26]. China's urban expansion is highly uneven between provinces and municipalities, and the "core–periphery" growth pattern of cities has exacerbated economic disparities between Chinese provinces and municipalities [27]. At present, different scholars have studied the relationship between urban expansion and economic development from different perspectives. One category is to measure the correlation between urban sprawl and economic development [28]; Zhang analyzed the relationship between urban sprawl and economic development using regression analysis. The results show that the level of economic development of a city is positively correlated with urban land expansion [29]. Xie explored whether there is an inevitable link between urban construction land expansion and economic growth using 108 prefecture-level cities in the Yangtze River Economic Belt, and the results show that urban expansion in the Yangtze River Economic Belt has a catalytic effect on its own economic development, as well as the development of its surroundings [30]. Another type of analysis is related to the perspective of the coupling and coordination of urban land use changes with population, land finance, economic growth, and ecological changes in the urbanization process [31–33]. Previous research is of great significance to recognize the relationship between urban expansion and economic growth, but the analysis perspective seldom considers the degree of separation between urban expansion and its own economic development and the spatial correlation with the economic development of surrounding regions.

China has a vast territory and significant regional differences, with different levels of urbanization in various regions [34–36]. As the fourth growth pole of China's economy, the Central Plains urban agglomeration has a medium level of urbanization in the country, and there is significant room for improvement in its urbanization level. Urban agglomeration is a spatial organization form of urban development, with closely related internal functions. Therefore, at the critical stage of its urbanization process, it is necessary to recognize the spatiotemporal evolution characteristics of urban expansion in the Central Plains urban agglomeration. Given the two-way causal relationship between urban expansion and economic development [37], it is of important practical significance to recognize the regional difference pattern of the correlation between urban expansion and economic development in order to promote high-quality urbanization and narrow the gap between urban expansion and economic development. Therefore, this study is based on vector data of urban built-up area scope, and the ESTDA model is used to explore the spatiotemporal interaction characteristics of urban expansion from both temporal and spatial dimensions in the Central Plains urban agglomeration. After that, using bivariate spatial autocorrelation and decoupling models, this paper explores the correlation between urban expansion and surrounding economic development in the Central Plains urban agglomeration, as well as the coordination with its own economic development. The results of this study have important theoretical and practical significance for clarifying the development and control objectives of urban expansion in the Central Plains urban agglomeration, providing important reference significance for formulating reasonable regional development policies for the Central Plains urban agglomeration, as well as providing new research perspectives and methods for the analysis and evaluation of the process of urban expansion in the academic community.

## 2. Materials and Methods

### 2.1. Study Area

The Central Plains urban agglomeration is one of the seven major urban agglomerations in China, located in the central-eastern part of China. According to the "Central Plains Urban Agglomeration Development Plan" released by the State Council in 2016, the Central Plains urban agglomeration includes 30 prefecture-level cities in 5 provinces, including Henan Province, Shanxi Province, Hebei Province, Shandong Province, and Anhui Province, as shown in Figure 1; among them, Zhengzhou City is the center and Luoyang is the subcenter. The total area is 287,000 square kilometers, the total population is about 158 million people, and the total GDP was about CNY 8 trillion and 100 billion in 2020. There is good natural endowment, the advantages of both the north and south, and the terrain is mainly plains and hills, with climate conditions in the transition zone of the north and south that constitute a good human living environment. The Central Plains urban agglomeration, with a lower level of economic development than the Yangtze River Delta Urban Agglomeration, the Pearl River Delta Urban Agglomeration, and the Beijing–Tianjin–Hebei Urban Agglomeration, is the fourth growth pole and an important power source of China's economic growth, with a good industrial base. According to the 2017 annual development report of the Central Plains urban agglomeration, the urbanization rate of the permanent population is 49.6%. According to the recent goal of the "Central Plains Urban Agglomeration Development Plan", the urbanization rate of the permanent population will exceed 56% by 2020. In terms of transportation, it is a must-go road to carry east and west and connect north and south. The Central Plains urban agglomeration is of strategic importance to the Central China and even the whole country in promoting urbanization and widening the space for economic development.

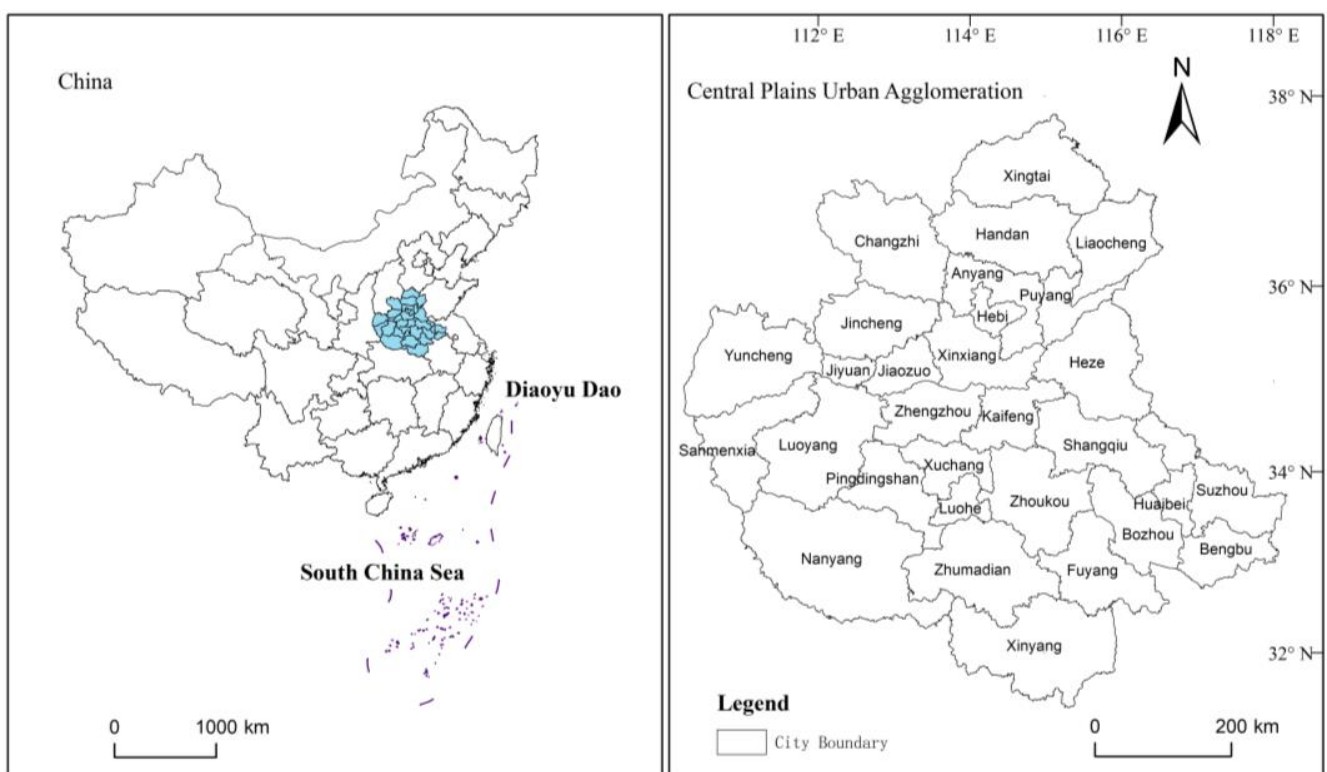

**Figure 1.** Location and study area: Central Plains urban agglomeration.

*2.2. Data Sources*

Most studies have used raster data or statistical yearbook data as the data source to analyze urban expansion. Given the limitations of the accuracy of these two types of data, vector data can portray the spatial pattern and actual area of urban built-up areas with higher accuracy. Therefore, this paper selects "A standardized dataset of built-up areas of China's cities with populations over 300,000 for the period 1990–2015 (https://www.doi.org/10.11922/sciencedb.j00076.00004 (accessed on 1 August 2022))" [38], as well as "A dataset of built-up areas of Chinese cities in 2020 (https://doi.org/10.11922/sciencedb.j00001.00332 (accessed on 1 August 2022))" [39]. This dataset is the vector data of urban built-up areas published by Science Data Bank and the Chinese urban dataset, developed with reference to the United Nations built-up area standard. It is produced with high-precision data sources, and the data products are highly accurate, with an average overall accuracy and kappa coefficient of 92% or more, meeting the current requirements of urban sustainable development research. As a result, they can be of greater use in applications such as urban expansion analysis and environmental assessment. Using the 2020 administrative division range as the standard, the original data were cropped and merged to form the vector data of the urban built-up area range of the Central Plains urban agglomeration for six periods. In order to maintain the accuracy of the area, the original data projection is uniformly converted to a Lambert equal area projection. Administrative division map vector data were obtained from the standard map service website (http://bzdt.ch.mnr.gov.cn/ (accessed on 1 August 2022)). GDP data for municipal districts for each year were obtained from the China City Statistical Yearbook for each year and were adjusted according to the administrative divisions in 2020. The GDP data of municipal districts in each year were obtained from the China City Statistical Yearbook of each year, and were grouped and adjusted according to the administrative divisions in 2020. In order to ensure the comparability of GDP in different years, it was uniformly adjusted to the real GDP, with 2005 as the base period, by calculating the GDP deflator.

*2.3. Methods*

2.3.1. Expansion Intensity Index

Expansion intensity is the ratio of urban built-up area expansion to the land area of a region within a certain time span, and it is often used to compare the strength, speed, and trend of urban built-up area expansion within a region in different periods [40]. The formula is

$$I = \Delta U / (\Delta t * TA) * 100\% \tag{1}$$

where I is the expansion intensity; TA is the total urban land area in the study area; $\Delta U$ is the expansion area of urban built-up area in different periods; and $\Delta t$ is the time interval. The larger I is, the greater the expansion intensity of urban built-up area in the characterized region; I can be used to compare the spatial differences of urban expansion in different spatial units.

2.3.2. Exploratory Spatiotemporal Data Analysis

Exploratory spatiotemporal data analysis (ESTDA) is a collection of spatial data analysis methods. It includes global spatial autocorrelation, local spatial autocorrelation, LISA spatiotemporal path, LISA spatiotemporal leap, etc. It is mainly used to analyze the spatial and temporal attribute data of geographic feature datasets and is a key research field of geographic information science (GIS) [41]. The ESTDA analysis will take into account spatial relevance based on geographical perspectives, as well as temporal factors. It overcomes the defect that traditional exploratory spatial analysis (ESDA) ignores the temporal element.

The Global Moran's I index was selected to identify the overall spatial pattern characteristics of the study area's urban sprawl with statistical significance [42]. The Global Moran's I index takes values between −1 and 1, with values less than 0 indicating negative spatial correlation, i.e., a spatial unit and its neighbors have similar "low–low" attributes clustered together, values equal to 0 indicating no correlation, i.e., an attribute value of a spatial unit is randomly distributed in space, and values greater than 0 indicating positive spatial correlation. In other words, the spatial unit and its neighbors have similar "high–high" attributes clustered together; the closer its value is to 0, the less concentrated the spatial distribution of a certain attribute of the spatial unit, and the closer its absolute value is to 1, the more obvious the spatial clustering of a certain attribute of the spatial unit.

In order to measure the local spatial correlation pattern, the local Getis–Ord Gi* index was selected in this study to identify urban sprawl hot and cold spots with statistical significance [43], i.e., it is used to identify the correlation degree of spatial location i, within a distance of d, with other spatial locations j. When the Z value is positive and statistically significant, it represents that the value within the region of distance d around cell i is greater than the regionwide expected value, presenting high value clustering, i.e., hot spot region; when the Z value is negative and passes the significance test, it represents that the value within the distance d region around cell i is less than the expected value of the whole region for low-value spatial clustering, i.e., cold spot region.

In order to explore the spatial correlation between the expansion of urban space and economic development, this study selects bivariate spatial autocorrelation to measure the spatial correlation between the two [44]. The global bivariate Moran's I can analyze the spatial correlation of the two attributes in the whole study area. If the bivariate autocorrelation coefficient is significantly positive, it indicates that the two attributes are spatially positively correlated and tend to be global; if the bivariate autocorrelation coefficient is significantly negative, it indicates that the two attributes are spatially negatively correlated and tend to be globally dispersed. Bivariate local Moran's I can analyze the local spatial correlation of the two attributes. If the local bivariate autocorrelation coefficient is significantly positive, it indicates that the two attributes are spatially positively correlated at spatial cell i. If the local bivariate autocorrelation coefficient is significantly negative, it indicates that the two attributes are spatially positively correlated at spatial unit I, which then indicates that the two attributes are spatially negatively correlated at spatial unit i.

LISA time path analysis, by calculating the geometric characteristics of coordinate movement trajectories of spatial units in Moran scatter diagrams, analyzes the spatially and temporally dependent interactive change characteristics of urban expansion intensity at local scales in a certain period, thus transforming spatially dependent characteristics into dynamic expressions. The main measures of LISA time path include relative length and curvature, which are calculated as follows [45]:

(1) Relative length of LISA time path

$$
Len_i = \frac{n * \sum\limits_{t=1}^{T-1} d(L_{i,t}, L_{i,t+1})}{\sum\limits_{i=1}^{n} \sum\limits_{t=1}^{T-1} d(L_{i,t}, L_{i,t+1})}
\tag{2}
$$

where $n$ is the number of research units; T is the annual time interval; $L_{i,t}$ is the LISA coordinate of the research unit in year $t$; and $d(L_{i,t}, L_{i,t+1})$ is the moving distance of research unit $i$ from year $t$ to year $t + 1$. The larger the $d$, the stronger the dynamic of the local spatial structure; $d > 1$ represents that the moving distance of research unit $i$ is greater than the average of the moving distance of research units.

(2) LISA time path curvature

$$
C_i = \frac{\sum\limits_{t=1}^{T-1} d(L_{i,t}, L_{i,t+1})}{d(L_{i,1}, L_{i,T})}
\tag{3}
$$

where $d(L_{i,1}, L_{i,t})$ is the moving distance of research unit $i$ from the first year to the last year. The larger $C_i$ is, the more curved the LISA time path is, and the more tortuous local spatial structure fluctuations are. If $C_i > 1$, it indicates that the moving tortuosity of research unit $i$ is higher than the average.

### 2.3.3. Standard Deviation Ellipse

The Standard Deviational Ellipse (SDE) model is a method for measuring the characteristics of the directional distribution of spatial data [46,47]. In the standard deviation ellipse, the long semiaxis represents the directional characteristics of the distribution of the data, and the short axis represents the range of the distribution of the data; the larger the difference between the long and short semiaxes, the more significant the directional distribution characteristics of the data, and the shorter the short semiaxis, the more pronounced the centripetal force of the data [48]. This study uses the standard deviation ellipse model to analyze the spatial party evolution characteristics of urban expansion in the Central Plains urban agglomeration in different periods, as well as the spatial distribution pattern characteristics. The three parameters of the standard deviation ellipse are the center of the circle, the angle of rotation, and the length of the *X*- and *Y*-axis, which are calculated as follows [49]:

(1) Center of ellipse:

$$
\begin{aligned}
SDE_x &= \sqrt{\frac{\sum\limits_{i=1}^{n} (x_i - \bar{x})^2}{n}} \\
SDE_y &= \sqrt{\frac{\sum\limits_{i=1}^{n} (y_i - \bar{y})^2}{n}}
\end{aligned}
\tag{4}
$$

(2) Rotation angle:

$$
\tan\theta = \frac{\left(\sum\limits_{i=1}^{n} \tilde{x}_i^2 - \sum\limits_{i=1}^{n} \tilde{y}_i^2\right) + \sqrt{\left(\sum\limits_{i=1}^{n} \tilde{x}_i^2 - \sum\limits_{i=1}^{n} \tilde{y}_i^2\right)^2 + 4\left(\sum\limits_{i=1}^{n} \tilde{x}_i \tilde{y}_i\right)^2}}{2\sum\limits_{i=1}^{n} \tilde{x}_i \tilde{y}_i}
\tag{5}
$$

(3) Standard deviation of *X*-axis and *Y*-axis:

$$\sigma^x = \sqrt{\frac{\sum\limits_{i=1}^{n}(\widetilde{x}_i\cos\theta-\widetilde{y}_i\sin\theta)^2}{n}}$$
$$\sigma^y = \sqrt{\frac{\sum\limits_{i=1}^{n}(\widetilde{x}_i\sin\theta-\widetilde{y}_i\cos\theta)^2}{n}} \tag{6}$$

where $x_i$ and $y_i$ represent the spatial coordinates of each element; $\overline{x}$ and $\overline{y}$ are the arithmetic mean center; $\widetilde{x}$ and $\widetilde{y}$ are the deviation of the *x* and *y* coordinates from the mean center; and *n* is the total number of elements.

### 2.3.4. Decoupling Model

The Tapio decoupling model is based on the decoupling elasticity to construct evaluation indicators; it was widely used because it was not affected by the error caused by the choice of the base period [50,51]. This paper constructs a decoupling elasticity model of urban expansion and economic growth based on the Tapio model to comprehensively analyze the disengagement of the relationship between urban spatial expansion and economic growth, where the urban built-up area change is used to denote the urban spatial expansion variable and the municipal GDP is used as the economic variable to construct the following decoupling model:

$$e = \frac{\Delta A / A}{\Delta GDP / GDP} \tag{7}$$

where e is the decoupling elasticity, and ΔA and ΔGDP are the amount of change in the built-up area of the municipal area and GDP at the beginning and end of the study, respectively. Based on the magnitude of the e value, with reference to the decoupling state set by Tapio, the decoupling state is specifically divided into three states and eight levels, and the specific classification criteria are listed in the following Table 1.

**Table 1.** Criteria for delineating the degree of decoupling between urban expansion and economic growth.

| Decoupling Type | | ΔA | ΔGDP | Decoupling Elasticity (e) |
|---|---|---|---|---|
| Negative decoupling | Growth negative decoupling | >0 | >0 | e > 1.2 |
| | Strong negative decoupling | >0 | <0 | e < 0 |
| | Weak negative decoupling | <0 | <0 | 0 < e < 0.8 |
| Decoupling | Weak decoupling | >0 | >0 | 0 < e < 0.8 |
| | Strong decoupling | <0 | >0 | e < 0 |
| | Recession decoupling | <0 | <0 | e > 1.2 |
| Connections | Growth Links | >0 | >0 | 0.8 < e < 1.2 |
| | Recession Link | <0 | <0 | 0.8 < e < 1.2 |

## 3. Results

### 3.1. Analysis of Urbanization Scale Change Characteristics

From 1990 to 2020, the urban built-up area of the Central Plains urban agglomeration expanded by 3159.06 km$^2$, with an average annual expansion of 105.3 km$^2$. As shown in Table 2, the total area of urban expansion in different periods showed a gentle "J"-type exponential growth, indicating that the urban expansion of the Central Plains urban

agglomeration showed a gradual upward trend and a state of active expansion. The total expansion area in the 2010–2015 and 2015–2020 periods exceeded 800 km$^2$, with a total expansion area of 1670.18 km$^2$, accounting for one half of the total expansion area. The expansion speed in both periods exceeded 160 km/a, a significant increase compared with 53.72 km/a in the 1990–1995 period, indicating that urban construction efforts have been significant in the past 10 years, with a significant expansion of the urban scope, and the overall regional urban construction is less subject to government regulation and control.

**Table 2.** Change in urban built-up area in Central Plains urban agglomeration.

| Index | 1990–1995 | 1995–2000 | 2000–2005 | 2005–2010 | 2010–2015 | 2015–2020 |
|---|---|---|---|---|---|---|
| Area growth (km$^2$) | 268.63 | 303.72 | 344.24 | 572.29 | 816.45 | 853.73 |
| Average annual expansion (km$^2$/year) | 53.726 | 60.744 | 68.848 | 114.458 | 163.29 | 170.746 |

The intensity of the expansion varies from different periods and shows clear phase-by-phase characteristics. As shown in Figure 2, it can be roughly divided into three stages, of which the 1990–2005 period is the first stage, during which the regional urban sprawl index grew more slowly, indicating that during this 15 years period, the whole area of the region grew more evenly and maintained a stable development trend. The period of 2005–2015 is the second stage, with a steep linear rise in the expansion intensity index, also reflecting the dramatic increase in the volume and rate of area growth during this period, during which the urbanization process accelerated. In the third stage in the 2015–2020 period, the expansion intensity still maintained an upward trend, but the increment slowed down. In recent years, the country has attached great importance to the construction of ecological civilization and urban human settlements, and it is not pursuing increments in urban expansion.

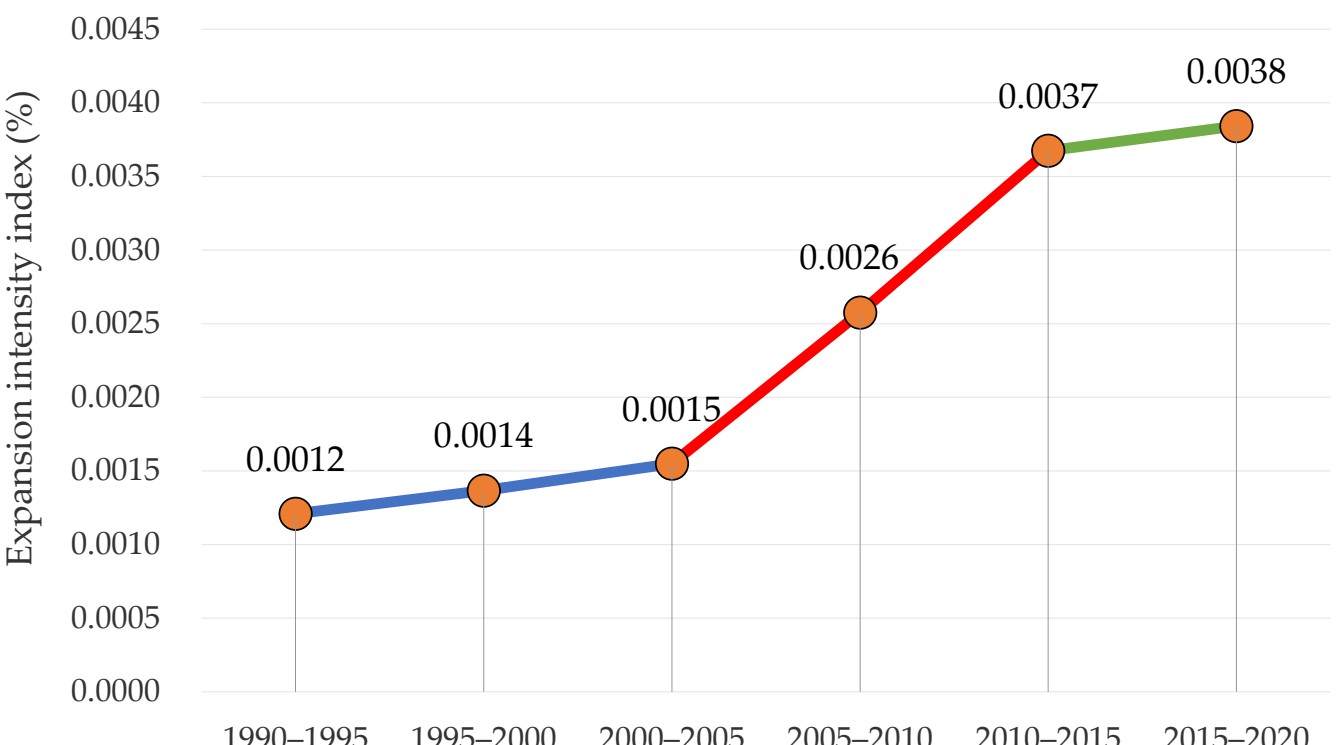

**Figure 2.** Expansion intensity Central Plains urban agglomeration in different periods.

As shown in Figure 3, from the city scale, the expansion intensity of Zhengzhou is increasing year by year, which reflects that the urbanization development of Zhengzhou, as

the core city of the Central Plains urban agglomeration, has been accelerating, and the level of urban expansion has been increasing. The regional development strategy of Zhengzhou of "expanding to the east, expanding to the west, extending to the south and linking to the north", as well as the planning of many new areas around Zhengzhou, have made the urban built-up area of Zhengzhou expand continuously. As a municipality directly under the central government of Henan Province, the intensity of urban expansion in Ji yuan has been at a high level compared with other cities. The built-up area of the city has increased from 3.67 km$^2$ in 1990 to 48.21 km$^2$ in 2020. Ji yuan has been adhering to the strategy of strengthening the city through industry in recent years. The development of industry has made considerable progress, which has led to employment, population concentration, and urban expansion. Luoyang, Anyang, Xinxiang, Pingdingshan, Jiaozuo Zhumadian, Puyang, and Jincheng have a moderate level of urban expansion. The rest of the cities have lower levels of expansion intensity, among which Xinyang, Sanmenxia, Changzhi, and Suzhou had the weakest expansion intensity during 30a, indicating that the urban expansion process in these regions is slower than other regions, and the regional development is at a lower level in the region.

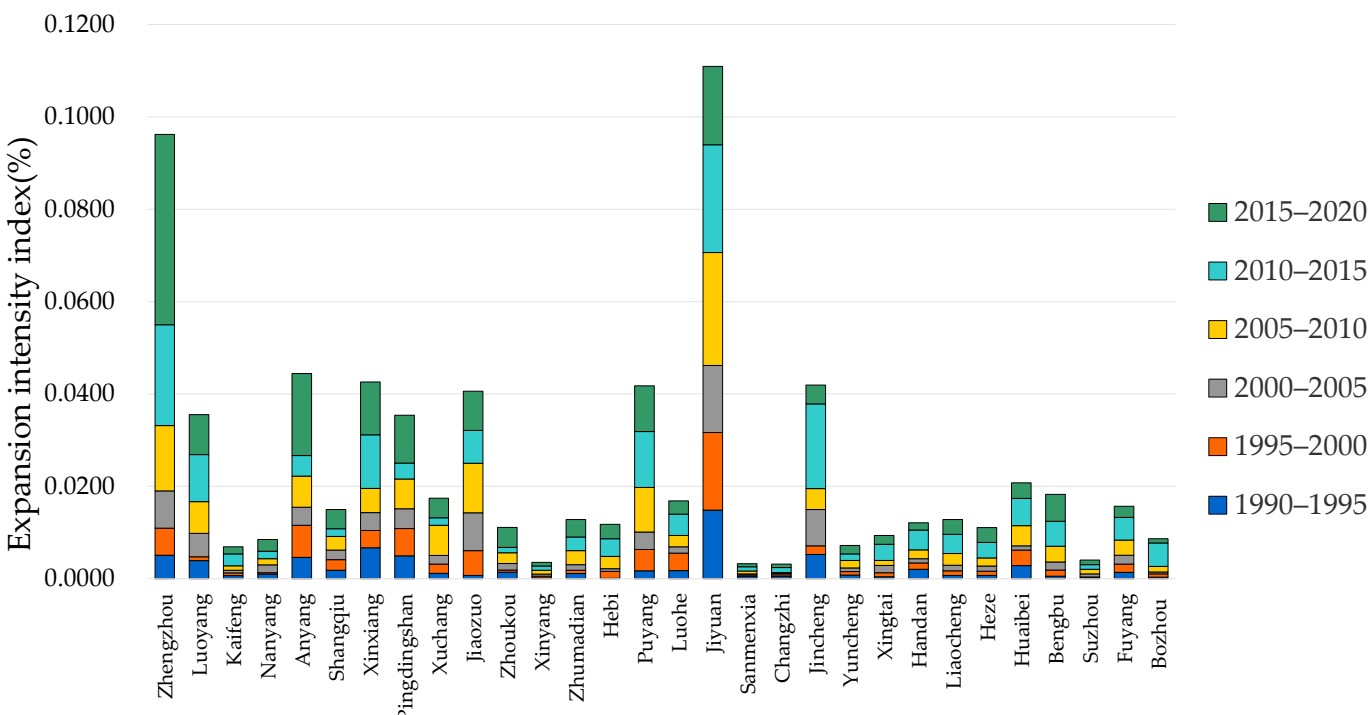

**Figure 3.** Expansion intensity of different cities in different periods.

### 3.2. Analysis on the Evolution Characteristics of Urbanization Direction

The standard deviation ellipse of urban built-up areas in each period was calculated by using the directional distribution tool in ArcGIS software to measure the directional distribution characteristics of urban areas in different periods. The calculation results are shown in Table 3 and Figure 4. Overall, the urban space of the Central Plains urban agglomeration has been mainly in the direction of "northwest–southeast" in the past 30 years. From 2010 to 2020, the elliptical azimuth angle changed from 148.93° to 125.5°; the direction of urban expansion is advancing eastward. After 2010, with the continuous optimization and upgrading of the internal structure of the urban system, resources have been used efficiently, and the economy, society, and environment of urban agglomerations have developed in a coordinated manner. Some cities will step into the new development stage, mainly present in the east extension and driven by Zhengzhou and Luoyang. From the perspective of ellipse oblateness, the ellipse oblateness gradually decreased from 0.25 to

0.16 from 1990 to 2010, and the directionality of urban space expansion became weaker and weaker. During this period, all regions are in a period of urgent need for development. Due to the gradual reduction in directionality brought by external forces, urban construction mainly assumes a regional "natural" growth. From 2010 to 2020, the oblateness rate gradually increased, from 0.16 to 0.35, and the direction gradually strengthened, mainly extending to Anhui Province. From 1990 to 2015, the total area of the ellipse gradually decreased from 1.23 to 0.98, indicating that the centripetal force of urban development was obvious during this period; the expansion of urban space was mainly in the form of "filling", and the urban land space was relatively concentrated. From 2015 to 2020, it increased slightly, showing the trend of "extension" in urban expansion.

**Table 3.** Standard deviation ellipse parameters of different periods.

| Year | 1990 | 1995 | 2000 | 2005 | 2010 | 2015 | 2020 |
|---|---|---|---|---|---|---|---|
| Area (km$^2$) | 1.23 | 1.12 | 1.06 | 1.05 | 1.06 | 0.98 | 1.10 |
| Elliptical azimuth | 125.3° | 136° | 141.30 | 143.31 | 148.93 | 119.96 | 125.50 |
| Elliptical oblateness | 0.25 | 0.18 | 0.20 | 0.17 | 0.16 | 0.20 | 0.35 |

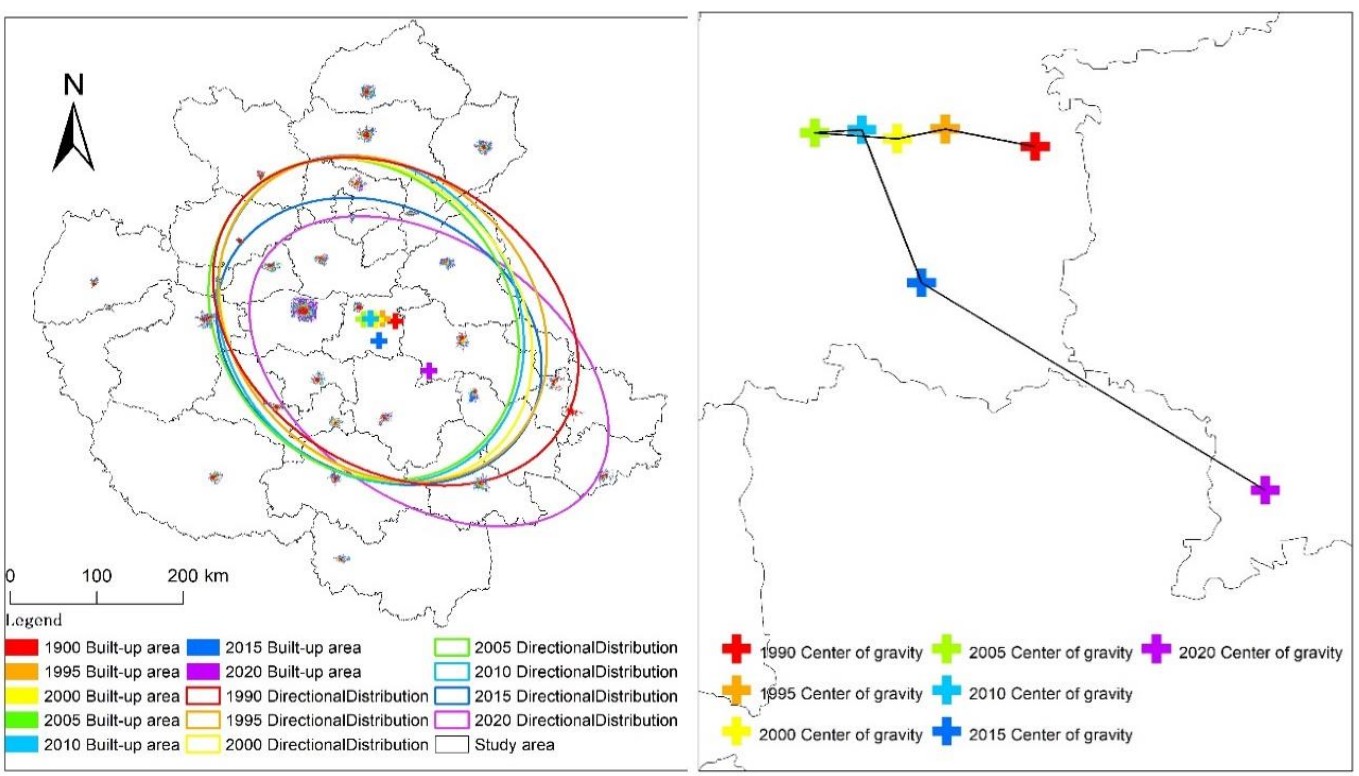

**Figure 4.** Standard deviation ellipse of urban built-up area and distribution of urban center of gravity.

Using the average center tool in ArcGIS 10.7 (ESRI Inc., Redlands, CA, USA), the urban center of gravity of the Central Plains urban agglomeration in each period was calculated, and the results are shown in Figure 4. From the perspective of the distribution of the center of gravity of the urban space, the center of gravity of the entire region hovered in Kaifeng from 1990 to 2010. Since 2010, the urban center of gravity has continuously moved in a southeast direction, and by 2020, the urban spatial center of gravity moved to Shangqiu. The direction of movement of the center of gravity of the city is relatively consistent with the development strategy of the entire region, i.e., "expansion to the east" and "extension to the south", which also reflects the effect of the regional development strategy.

*3.3. Analysis of Spatiotemporal Interaction Characteristics of Urbanization*

3.3.1. Analysis of Spatial Agglomeration Characteristics

The expansion intensity index I is selected as the measurement index, and the spatial agglomeration characteristics of urban expansion in the study area are measured with the help of the Global Moran's I index. The results show that the Moran's I index for the 1990–1995 and 1995–2000 periods were 0.03 and 0.02, respectively, and the *p* value was greater than 0.05, indicating that the level of urban expansion in these two periods was characterized by a random spatial distribution, and the development interaction between cities was relatively small. In the 2000–2005, 2005–2010, and 2010–2015 periods, the Moran's I indexes were 0.34, 0.18, and 0.17, respectively, and the significance levels were 0.1, 0.5, and 0.5, respectively, reflecting that the intensity of urban expansion during this period was characterized by spatial agglomeration, and the agglomeration characteristics reflected a gradually decreasing trend, indicating that the development of the city in this period was influenced by the neighboring cities. The Moran's I index from 2015 to 2020 is about 0.01, and it has not passed the significance test. The intensity of urban expansion is randomly distributed in space.

Taking the intensity of urban expansion in different periods as the measurement index, with the help of spatial statistical analysis tools provided by Geoda software, the Ge-tis–Ord Gi* index for six periods was calculated, and the evolution map of urban expansion hotspots of the Central Plains urban agglomeration in different periods was drawn. As shown in Figure 5, the spatial distribution of hot spots is relatively stable in different periods of time, mainly in Jiaozuo and surrounding cities, showing a clustering trend. Zhengzhou, Luoyang, Kaifeng, Jiyuan, Xinxiang, and other cities around Jiaozuo have developed rapidly in recent years, with large-scale urban expansion, and the urban area increment is at a relatively high level in the region, Showing obvious "high–high" agglomeration characteristics, it has always been a hot spot of urban expansion in the Central Plains urban agglomeration. The spatial distribution of cold spots is characterized by instability. Since 1995, cold spot areas have appeared, among which Shangqiu has been in the cold spot area. It reflects the characteristics of the "low–low" agglomeration of the expansion intensity of Shangqiu and surrounding cities. On the whole, the distribution of cold and hot spots is generally characterized by a punctiform distribution of "hot in the northwest and cold in the southeast", but there are differences in the distribution form and number of hot and cold spots in different periods. Zhengzhou and Luoyang, as the center and subcenter of the urban agglomeration in the Central Plains, have obvious radiating effects on the surrounding areas, and the urban expansion level of the surrounding cities is relatively high. Although cities such as Shangqiu, Huaibei, Bozhou, Suzhou, and Bengbu in the southeast have maintained a trend of increasing urban expansion in recent years, the rate of increase has not been large. These cities are far from central cities such as Hefei, Zhengzhou, and Wuhan in the central region. Therefore, they mainly rely on their own internal development, and they are less affected by the surrounding areas. The level of urban expansion is relatively low in the region. During the study period, the number of hot and cold spots also fluctuated, with no obvious trend characteristics. From the perspective of changes in hot and cold spots, Jiaozuo was always in the hot spot during the study period, and Shangqiu has been in the cold spot since 1995.

3.3.2. Spatiotemporal Interaction Feature Analysis

Based on the above, the period from 1990 to 2020 is divided into a period of five years with a total of six periods, corresponding to the national economic and social development planning from the "Eighth Five-Year Plan" to the "Thirteenth Five-Year Plan". The Moran scatter diagram of the urban expansion index in the first and sixth periods was calculated with the help of Geoda software, and the spatial coordinates of each research unit in the Moran scatter diagram in different periods was obtained. The LISA time path length, curvature, and path movement direction were calculated according to Formula (2) and

Formula (3), and the local spatiotemporal interaction characteristics of the urban expansion intensity of the Central Plains urban agglomeration were analyzed.

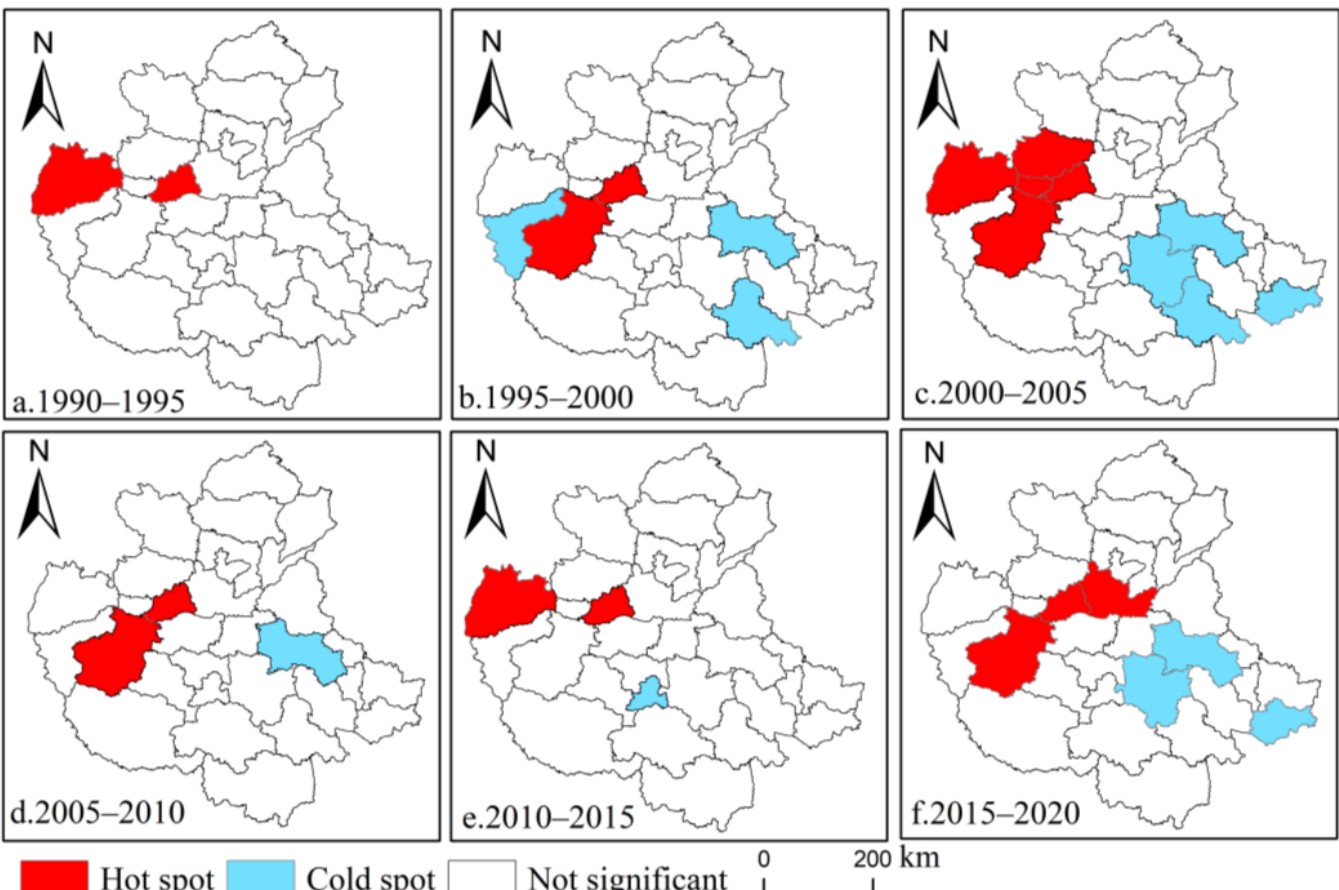

**Figure 5.** Evolution of hot and cold spots of urban spatial expansion. (**a**) 1990–1995 hot and cold spots of urban spatial expansion; (**b**) 1995–2000 hot and cold spots of urban spatial expansion; (**c**) 2000–2005 hot and cold spots of urban spatial expansion; (**d**) 2005–2010 hot and cold spots of urban spatial expansion; (**e**) 2010–2015 hot and cold spots of urban spatial expansion; (**f**) 2015–2020 hot and cold spots of urban spatial expansion.

The relative length of the LISA time path can reveal the local spatial dependencies and the spatiotemporal evolution characteristics of the spatial structure. The relative path length is divided into five levels by using the natural breakpoint method, and the results are shown in Figure 6. The relative length of urban expansion intensity in the Central Plains urban agglomeration generally presents a spatial distribution characteristic of low in the northwest and high in the southeast. This indicates that urban expansion in the northwest region has a more stable local spatial structure, while urban expansion in the southeast region has a more dynamic local spatial structure. The regions with relatively large lengths mainly include Xuchang, Jincheng, Liaocheng, Puyang, Fuyang, and Bengbu, indicating that these regions have a more dynamic spatial structure, and regional development policies should focus on steadily promoting the process of urban expansion.

The curvature of the LISA time path can reflect the size of the regional urban expansion affected by the neighborhood. Using the natural breakpoint method, the curvature is divided into five levels. The curvature in general presents a spatial distribution characteristic of high in the middle and low in the surrounding area, with an average value greater than 1, indicating that the urban expansion of the Central Plains urban agglomeration has a strong spatial dependence. Cities with larger curvature include Zhumadian, Pingdingshan, Luoyang, Anyang, Hebi, Xinxiang, Kaifeng, and Bengbu, indicating that

these cities have greater fluctuations in spatial dependence and are greatly affected by neighborhoods; except for the Bengbu accident, the rest of the cities are distributed on both sides of Zhengzhou, which also shows that the central megacities have a significant effect on the surrounding cities. Therefore, these cities need to strengthen their own development, enhance the resilience of urban development, and gradually form a relatively stable spatial relationship with the surrounding areas.

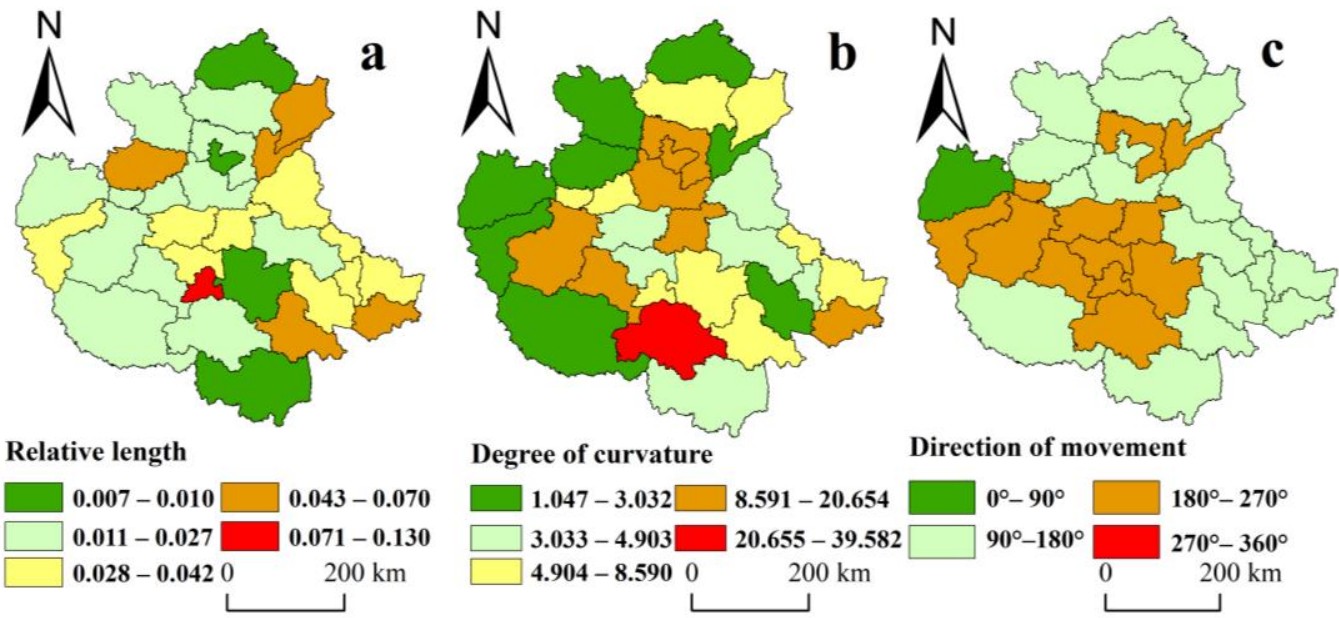

**Figure 6.** Schematic diagram of LISA time path analysis results. (**a**) Relative length of LISA time path; (**b**) LISA time path curvature; (**c**) LISA time path moving direction.

According to the coordinate position and moving trajectory of the urban expansion intensity of different cities in different periods in the Moran's I scatter diagram, the migration direction of the LISA path is calculated, and the moving direction of the LISA coordinates of the research unit is divided into four quadrants. The first quadrant is 0–90°, indicating that the research unit and the neighborhood show a "high–high" growth trend; 90~180° is the second quadrant, indicating that the research unit and its neighborhood are in a "low–high" growth trend, the research unit itself has a low growth rate, and the neighborhood has a high growth rate; 180~270° is the third quadrant, showing a "low–low" growth trend. In a "low–low" growth trend, the research unit and its neighbors show a negative synergistic growth. The fourth quadrant is 270~360°, which is a "high–low" growth trend: the urban area unit and its neighboring cities show a reverse growth, and the research unit itself has a high growth rate, indicating neighborhood low growth. As shown in Figure 5, there are 12 cities showing a "low–high" growth trend, accounting for 40%, and they are mainly distributed in the northwest and southeast directions centered on Xuchang in space, showing a state of spatial agglomeration. There are 17 cities showing a "low–low" growth trend, accounting for 57%, and the spatial distribution has the characteristics of agglomeration, concentrated in the periphery of the Central Plains urban agglomeration.

*3.4. Analysis of the Correlation between Urban Spatial Expansion and Economic Development*

3.4.1. Analysis of the Correlation between Urban Expansion and Peripheral Economic Development

According to the above, the urban expansion speed and expansion intensity of the Central Plains urban agglomeration have greatly increased since 2005 and have entered a stage of rapid development. The three periods of "2005–2010", "2010–2015", and "2015–2020" discuss the spatial coordination between rapid urban expansion and economic development. GDP is an important indicator of a region's economic development, and it represents

the level of regional economy to a certain extent. Therefore, this paper uses the GDP of municipal districts to represent the economic development level of different research units. According to the bivariate global autocorrelation model, the change rate of urban built-up area in different stages is selected as the first variable and the change rate of GDP is the second variable, and the spatial agglomeration characteristics of urban expansion and economic development of the Central Plains urban agglomeration in different periods are analyzed with the help of Geoda software. The results show that the global Moran's I index of the two variables in the three stages is negative and has not passed the significance test, indicating that from a global perspective, urban expansion and the economic development of surrounding cities do not exhibit spatial agglomeration characteristics.

This study uses the local bivariate Moran's I index to further explore the spatial agglomeration degree and agglomeration area of urban expansion and economic development in the Central Plains urban agglomeration. As shown in Figure 7, four aggregation types are divided, namely "high–high", "high–low", "low–high", and "low–low". It can be seen from Figure 7 that the local clustering types of the two variables are different in different periods, and the local clustering characteristics are not significant in most regions. The "high–high" agglomeration feature only appeared in Kaifeng from 2010 to 2015. This indicates that the high level of urban expansion in this region and the high level of economic development in the surrounding cities have gathered, and that urban expansion and economic development in the surrounding cities have shown coordinated growth. The "low–low" type of agglomeration area indicates that there is a positive correlation between low urban expansion and the low economic development level of surrounding cities. This type of area occurred in Handan from the 2005–2010 to 2015–2020 periods, indicating that the low change rate of urban area in Handan and the low level of economic development in the surrounding areas appeared as an agglomeration phenomenon during these two periods. However, from 2010 to 2015, Handan was characterized by "high–low" agglomeration, indicating that the urban construction of Handan was accelerating during this period, but the speed of economic development in the surrounding area was low, and the factors of urban expansion came more from its own government intervention; it was less affected by the surrounding economic development. During the "Eleventh Five-Year Plan" and "Twelfth Five-Year Plan" periods, Liaocheng was in a "high–low" agglomeration type, indicating that its own urban construction was not in harmony with the surrounding economic development. The "low–high" agglomeration areas appeared in Shangqiu from 2010 to 2015, and in Yuncheng and Huaibei from 2015 to 2020, indicating that the urban areas in these areas had a low increase in urban area, but the surrounding economic development has greatly improved. The two positively correlated agglomeration types of "high–high" and "low–low" in each period appeared less and did not show an increasing trend, indicating that the urban expansion of the Central Plains urban agglomeration has a low correlation with the economic development of surrounding cities.

### 3.4.2. Analysis of Decoupling between Urban Expansion and Its Own Economic Development

Using the Tapio decoupling model, the decoupling index of 30 prefecture-level cities in the Central Plains urban agglomeration was calculated. Due to the spatial lag effect of urban expansion and economic growth, the three periods of 2005–2010, 2010–2015, and 2015–2020 were selected for decoupling calculations. According to the classification criteria of decoupling types, the decoupling elasticity of urban expansion and economic development in different stages was discussed, as shown in Figure 8. The types of decoupling between urban expansion and economic growth of the Central Plains urban agglomeration mainly appear in three stages: negative growth decoupling, weak decoupling, and growth connection, but the decoupling types show obvious changes in quantity and spatial distribution in different periods. The number of cities with growth negative decoupling and growth connection decoupling has increased, and they are clustered towards the re-

gional center. Weak decoupling is the main type of decoupling at each stage, but there is a tendency to gradually decrease.

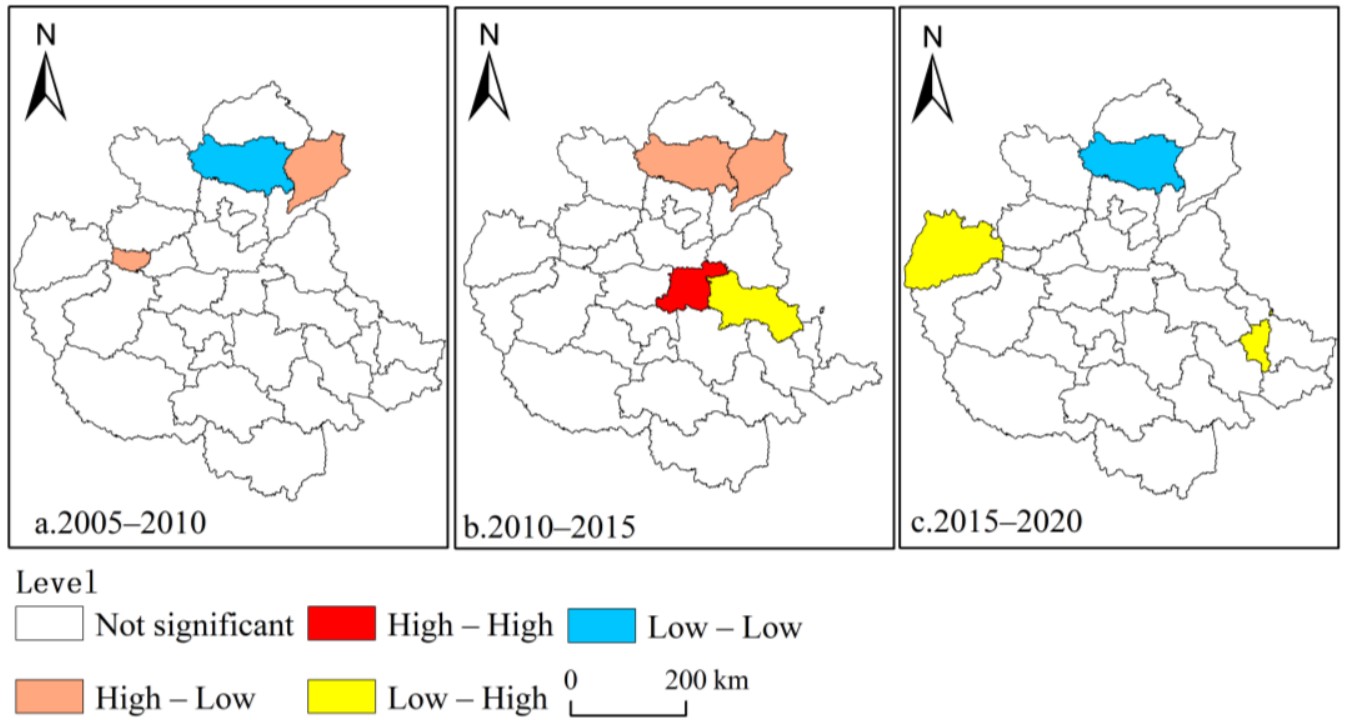

**Figure 7.** Spatial correlation between urban expansion and economic growth. (**a**) 2005–2010 Local bivariate spatial autocorrelation; (**b**) 2010–2015 Local bivariate spatial autocorrelation; (**c**) 2015–2020 Local bivariate spatial autocorrelation.

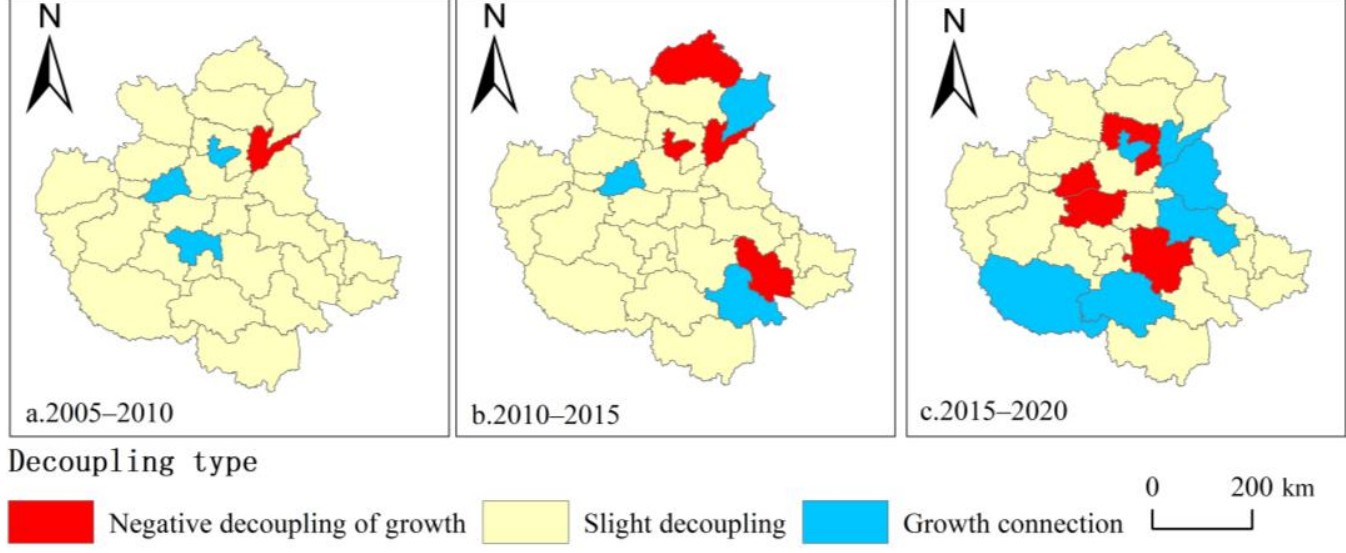

**Figure 8.** Types of decoupling between urban expansion and economic growth. (**a**) 2005–2010 Decoupling type; (**b**) 2010–2015 Decoupling type; (**c**) 2015–2020 Decoupling type.

Specifically, there was only one negative decoupling of growth in the first stage, which appeared in Puyang, and three types of growth connections appeared in Hebi, Jiaozuo, and Xuchang. The negative decoupling of growth and the connection of growth reflect the imperfect coupling between the expansion of these cities and economic development; that is, the expansion of cities does not bring about corresponding economic growth. The rest of

the cities are all weakly decoupled, accounting for 86.7% of the research samples (Table 4), reflecting a slight inconsistency between the expansion of these cities and economic development. There is a certain degree of dependence between economic development and land consumption, but the degree of dependence is not high. In the second stage, negative growth decoupling increased, accounting for 13%, the proportion of weak decoupling decreased by 10%, and the proportion of growth connections remained unchanged. The cities with negative growth decoupling in terms of spatial distribution included Puyang, Hebi, Xingtai, and Bozhou, and growth connections appeared in Jiaozuo, Fuyang, and Liaocheng. The proportion of negative growth decoupling in the third stage has not changed compared with the previous stage. The proportion of weak decoupling continues to decrease by 10%, and the proportion of growth connection increases by 10%. The spatial distribution shows a big change. Among them, cities with negative growth decoupling include Anyang, Jiaozuo, Zhengzhou, and Zhoukou, and cities with growth connections include Hebi, Puyang, Heze, Shangqiu, Zhumadian, and Nanyang.

**Table 4.** Number (pcs) and proportion of different decoupling types in each period.

| Stage | Growth Negative Decoupling | Weak Decoupling | Growth Connection |
|---|---|---|---|
| First (2005–2010) | 1/3.3% | 26/86.7% | 3/10% |
| Second (2010–2015) | 4/13.3% | 23/76.67% | 3/10% |
| Third (2015–2020) | 4/13.3% | 20/66.67% | 6/20% |

Overall, weak decoupling is the main feature of the relationship between urban expansion and economic growth in the three periods, but it also shows a trend of increasing from weak decoupling to negative growth decoupling and growth linkage. Although the urban expansion and economic development of the Central Plains urban agglomeration have been greatly improved from the "Eleventh Five-Year Plan" to the "Thirteenth Five-Year Plan", the uncoordinated trend between urban expansion and economic growth will continue in the future. During the "Fourteenth Five-Year Plan" period, more stringent requirements were put forward for coordinating the relationship between urban expansion and economic development, and this incoordination appeared to gather around Zhoukou in the southeast. According to the above, in recent years, the direction and center of urban expansion in the Central Plains urban agglomeration have migrated to the southeast. Therefore, relevant departments should pay attention to the process of urban expansion and should also focus on the coordination of economic growth and land resource consumption to avoid the phenomenon of uneconomical consumption of land resources occurring in the middle of the country, so that the regional urban construction and economic development can be coordinated and steadily advanced. The slowdown in the expansion of construction land and the steady and healthy development of economic growth are the essence of decoupling performance.

## 4. Discussion

Traditionally, the analysis of spatiotemporal characteristics of urban expansion mainly uses statistical data on urban construction land or uses single remote sensing image data to interpret the scope of the built-up area. Although using these two types of data has certain feasibility, there are also certain limitations. Statistical data often have large errors and lack spatial location information, which can only be analyzed from the perspective of mathematical statistics. Using remote sensing image interpretation by researchers, it is difficult to obtain high-precision remote sensing images, and obtaining long urban built-up area range time series data while ensuring accuracy is a significant workload. This study adopts the vector data of high-accuracy and long time series of urban built-up areas published by scholars, providing convenience and data accuracy for this study. With the progress of science and technology, more and more scholars are trying to explore the study of urban using multisource data [52,53]. The further exploration direction of this study is to comprehensively identify the inherent laws and characteristics of urban development from

different aspects using multisource data. In terms of research perspective, traditionally, the measurement perspective of urban expansion mainly focuses on the characteristics of expansion scale and spatial pattern. The analysis of its evolution characteristics over time often separates time and space, ignoring the analysis of spatiotemporal interaction characteristics. This study introduces the LISA time path analysis method, which dynamically analyzes the spatiotemporal pattern interaction characteristics of urban expansion process and has certain innovations. Traditionally, analyzing the relationship between urban expansion and economic growth has focused only on the correlation analysis between urban expansion and its own economic growth, ignoring the linkage analysis between urban expansion and surrounding economic development. This study applies the bivariate spatial autocorrelation analysis method to the spatial correlation analysis between urban expansion and surrounding economic growth, which has certain applicability. In considering the coordination between urban expansion and its own economic development, this study introduces a decoupling model to analyze the decoupling effect of urban expansion and economic growth more intuitively. However, in terms of horizontal economic development level, only GDP statistics were used. In subsequent studies, objective data such as nighttime lighting should be added to comprehensively measure economic development [54], making the evaluation of economic development more objective. Due to the interaction between urban expansion and ecological environment, industrial development, and population agglomeration [5,32,55–57], in subsequent research, the decoupling effect analysis between ecological effects, industrial development, population flow, and urban expansion will be considered, and the driving mechanism of urban expansion will be further analyzed to provide scientific suggestions for regional urban expansion and economic development, as well as improve the research results system of urban geography.

## 5. Conclusions

The urbanization level of the Central Plains urban agglomeration is currently in a rapid improvement stage, with urban expansion being an important component of urbanization. All economic and social development relies on land. In order to avoid the related negative impact of urban sprawl in the Central Plains urban agglomeration, it is necessary to explore the characteristics and laws of urban expansion in the Central Plains urban agglomeration in a timely manner. Currently, research on urban expansion in the Central Plains urban agglomeration ignores the discussion of spatiotemporal interaction characteristics. This study analyzes the scale characteristics, direction characteristics, and spatiotemporal interaction characteristics of urban expansion in the Central Plains urban agglomeration in the past 30 years based on more accurate long-term urban built-up area vector data, and it analyzes the spatial correlation and decoupling effect between the two in combination with economic statistics. This study provides a reference for relevant government management departments to judge the macro pattern of regional development and formulate policies, and it provides new ideas and methods for the study of urban expansion. The specific conclusions are as follows:

(1) The total area of urban built-up areas in the Central Plains urban agglomeration has shown a trend of increasing year by year, and the total expansion area in the past 10 years accounts for half of the total expansion area. The expansion intensity showed an "S"-shaped growth trend, with a slow increase in expansion intensity from 1990 to 2005, a rapid increase in expansion intensity from 2005 to 2015, and a steady trend in urban expansion intensity from 2015 to 2020. There are significant differences in expansion intensity among different cities. Zhengzhou has maintained a high intensity of expansion; cities with moderate expansion intensity are mostly located in Henan Province and are mostly distributed around Zhengzhou. Xinyang, Sanmenxia, Changzhi, and Suzhou had the lowest expansion intensity during the past 30 years. Therefore, in the process of urbanization, the Central Plains urban agglomeration should pay more attention to the problems caused by the strong expansion of Zhengzhou and its surrounding cities, and

at the same time, strengthen the cities with weak urban construction efforts to achieve coordinated regional development and narrow the gap.

(2) From 1990 to 2020, the urban spatial expansion of the Central Plains urban agglomeration mainly presented a "northwest–southeast" trend. The directionality of urban expansion first decreased and then increased, mainly extending towards Anhui Province. Urban space has shifted from primarily "infill" expansion in the early stage to primarily "extensional" expansion. From 2010 to 2020, the center of gravity of urban space shifted significantly to the southeast, moving from Kaifeng to Shangqiu. Due to the influence of the natural geographical environment, the development space in the west and southwest of the Central Plains Urban Agglomeration is limited. Driven by the radiation centered around Zhengzhou, due to superior terrain conditions, convenient transportation, and good industrial foundation, the southeast cities have continuously expanded their urban spatial scope in recent years, making the urban center of the Central Plains Urban Agglomeration constantly move towards the southeast. Kaifeng, Xuchang, and Shangqiu have become an important gateway for the urbanization of the Central Plains urban agglomeration to the southeast.

(3) The spatial agglomeration characteristics of urban expansion in the Central Plains urban agglomeration are weak. The hot spots for urban expansion mainly change in Jiaozuo and surrounding areas. Urban expansion in the Central Plains urban agglomeration has a strong spatial dependence, and cities in the southeast region have a more dynamic spatial structure. Urban expansion in the Central Plains urban agglomeration mainly reflects the coordinated low growth of peripheral cities.

(4) From 2005 to 2020, the spatial correlation between urban expansion of the Central Plains urban agglomeration and surrounding economic development was not significant, and there was a risk of decoupling between urban spatial expansion and its own economic growth. It is suggested that the urban expansion of the Central Plains urban agglomeration is mainly influenced by its own development policies and is less driven by the development of its surrounding areas. Moreover, there is a trend of increasing the risk of uncoordinated urban expansion and economic growth. It is recommended that the formulation of future regional development strategies should consider the coordination between urban expansion and economic growth, avoid blind expansion, and focus on the driving role of cities with high urbanization levels in the development of low urbanization levels.

In summary, this study analyzes the laws of urban expansion in the Central Plains urban agglomeration from the perspective of spatiotemporal dynamics, and it analyzes the correlation between urban expansion and internal and external economic growth, which better explains the spatiotemporal characteristics of urban expansion in the Central Plains urban agglomeration in the past 30 years, based on the decoupling risk of economic development. These results can provide reference for the future development planning of the Central Plains urban agglomeration. However, this study does not explain the mechanism of the spatiotemporal characteristics of urbanization in Central Plains urban agglomeration. In the future, further analyzing the driving factors that generate the spatiotemporal characteristics of urbanization in Central Plains urban agglomeration in combination with multisource data will be considered.

**Author Contributions:** Conceptualization, Z.W.; methodology, Z.W.; software, Z.W.; validation, Z.W. and L.W.; formal analysis, Z.W.; investigation, Z.W.; resources, Z.W.; data curation, Z.W.; writing—original draft preparation, Z.W.; writing—review and editing, Z.W. and Q.P.; visualization, Z.W.; supervision, Z.W.; project administration, L.W.; funding acquisition, B.Z. All authors have read and agreed to the published version of the manuscript.

**Funding:** This research was funded by the National Natural Science Foundation of China (grant number 42001129).

**Data Availability Statement:** The urban built-up area dataset from 1990 to 2015 is available at (http://www.doi.org/10.11922/sciencedb.j00076.00004), accessed on 1 August 2022. The urban built-up area dataset from 2020 is available at (https://doi.org/10.11922/sciencedb.j00001.00332),

accessed on 1 August 2022. The rest of the data presented in this study are available on request from the corresponding author.

**Acknowledgments:** We thank the relevant teams and organizations for providing the datasets used in this study.

**Conflicts of Interest:** The authors declare no conflict of interest.

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
