# Peer review of "Analysis of Spatiotemporal Interaction Characteristics and Decoupling Effects of Urban Expansion in the Central Plains Urban Agglomeration"

_land, doi:10.3390/land12040772_

Round 1
Reviewer 1 Report
Urbanization is an inevitable choice for socio-economic development, and it is important to investigate the spatio-temporal characteristics of cities and their coordination with economic growth. The study uses h ESTDA, standard deviation ellipse and decoupling models to analyze the spatio-temporal interaction characteristics and decoupling effects of urbanization in the Central Plains urban agglomeration, which provides a better idea for urban development research. However, there are still shortcomings in the article, as follows.
1. The research question of the article cannot be found in the introduction ; the introduction should clearly point out the research problem and innovation.
2. The title, lines 16-17 and 50-51, express the research object as urbanization and land urbanization respectively, which have different concepts. It is suggested that the authors should accurately define the research object and clarify the meaning of urbanization.
3. Lines 54-57, which describe spatio-temporal patterns and spatio-temporal characteristics, respectively, both of which are essentially the same content. It is recommended that the literature be reorganized to emphasize studies that are relevant to the content of this study.
4. Among the references, some of them are old, and it is suggested to supplement the studies in recent years.
5. At the end of the introduction, it is suggested to increase the article structure organization.
6. Lines 130-147, in the study area section, it is recommended to add a description of the urbanization of the area.
7. The research method is not simply a conceptual description, but should be elaborated in conjunction with the article research, reflecting the applicability of the method in this study and the necessity of selecting the method.
8. It is pointed out in the introduction that the use of vector data will improve data accuracy, and it is one of the innovative points, but only simple statistical analysis of data is made in the research results, which fails to reflect the advantages of using vector data. In addition, there are relatively more studies analyzing urbanization based on vector data, and it is not convincing enough to take it as an innovative .
9. It is pointed out in the introduction that the research content is land urbanization, but the existing research results only analyze the increase or decrease of construction land, and other land uses are not considered. At the same time, the study emphasizes the interaction of urbanization, but the study only analyzes the spatial distribution characteristics of cities and fails to reflect the interaction relationship in the urbanization process. It is suggested that the research object should be clarified, and the land use transfer matrix analysis and other contents can be added.
10. The elaboration of the coordinated relationship between economic development and urbanization needs to be further deepened, and the existing explanation is insufficient.
11. The degree of specification of the figure production should be further improved, for example, the position of the north compass placed in Figure 5-8. A scale can be placed in Figure 6.
12. The discussion is not a deficiency of the research study. It is recommended to add a discussion of the research results and compare them with previous studies to guide and support your research results.
13.Conclusions is not a simple description of the research results, and it is recommended to further deepen and refine them.
Author Response
Please view the attachment.

Reviewer 2 Report
This article analyzes the spatio-temporal interaction characteristics and decoupling effects of urbanization in the Central Plains urban agglomeration of China, using urban built-up area extent and GDP data from 1990 to 2020, combined with ESTDA, standard deviation ellipse, and decoupling models. The authors find that the urban built-up area is growing in a "J" shape, expanding mainly in the direction of "northwest-southeast", and showing local spatial structure instability and strong spatial characteristics. Furthermore, during 2005-2020, the number of cities with negative growth decoupling and growth linkage increased, and the number of weakly decoupled cities tended to gradually decrease, indicating the increasing risk of incompatibility between urban expansion and economic growth. However, the study does not consider the decoupling effects of ecological effects, industrial development, and population mobility from land urbanization.The article provides a new reference basis for the designation of regional development strategies, and is of great significance for the Central Plains Urban Agglomeration to promote new urbanization. I congratulate the author(s) on their effort to establish such a logical and original method. But I think there are some points in the work which need to be revised. I will try to explain them in the following paragraphes:
-Contribution to academia needs to be highlighted in the abstract, introduction and conclusion part of the study. The contribution of the study needs to be explained in such a way that to increase the originality of the study.
- Abstract should cover Introduction and Reason for conducting the research, the Problem (knowledge gap), Methods, Outcomes (results), and Ramifications (Implications). The abstract should be re-written so that it encompasses summaries of the most important parts of the study results and authors' arguments.
-In order to increase the internal validity of the study I am highly suggesting to cite the following articles: - The causal relationship between urbanization and economic growth in us: fresh evidence from the toda–yamamoto approach, urbanization, housing quality and health: towards a redirection for housing provision in nigeria, the phenomenon of mobility, a development challenge for the city of algiers
-The thesis statement should come at the end of the introduction.
- The authors may need to justify why this case study (s) and how the findings can be generalizable.
- For this empirical research, the results are not clearly presented.
- All the cited references should be directly relevant to the research. There are some articles in the text which doesn’t have a relation to the written text.
-Suggestion for future study is also missing from the last line of the conclusion. It should be used to point out any important shortcomings of the manuscript, which could be addressed by further research or to indicate directions for further work could take.
Author Response
Please view the attachment.

Reviewer 3 Report
This is a very interesting paper. Due to this I agree with the importance of this manuscript.
Regarding the possibilities to improve it, I would like to comment the following if it could help to the authors:
1. To highlight in the abstract and in the introductions the goal of the study, its value and the benefits of the results obtained.
2. To introduce more explications for each figure and table presented in the results section to understand them better than now.
3. To maintain a similar form of the presentation of the figures shown, when it is possible to look more similar configuration, figures and tables titles in the middle, etc.
4. To increase the comments of the section 4, it is very short comparing the other sections, introducing a bit more information about the discussions of the results with the references shown.
5. There is only 2 references of this year 2023. Please introduce more recent references increasing the number of references at least over 50.
Thanks so much and good job. I like this very much.
Author Response
Please view the attachment.
